# Twins: Learn to Predict Unified Representations with Focal Loss

Kaixiong Gong [* 1 2]  Xin Cai [* 1 2]  Bin Lin [2]  Hao Wang [3]  Yunlong Lin [4]  Mingzhe Zheng [2]  Bohao Li [5]
Jian-Wei Zhang [† 2]  Miles Yang [2]  Zhao Zhong [2]  Liefeng Bo [2]  Xiangyu Yue [1]

## Abstract

Unified multimodal models seek a shared visual token space that supports both multimodal understanding and image generation. Discrete methods unify the interface via a shared codebook, whereas continuous pipelines often rely on two disparate representations—semantic features (*e.g.*, ViT) for understanding and low-level latents (*e.g.*, VAE) for synthesis—resulting in mismatched latent spaces. We propose *Twins*, a unified continuous token space formed by channel-wise concatenating ViT and VAE features on the same token grid, so the sequence length is unchanged and attention cost does not increase. However, jointly modeling Twins in a Diffusion Transformer exposes a severe *optimization imbalance*: the model fits the ViT component well but struggles to match the VAE latent distribution. We trace this imbalance to three sources of heterogeneity: frequency bias, intrinsic dimensionality, and condition-aligned vs condition-independent uncertainty. To address it, we adapt a focal regression objective for flow matching that upweights large-error VAE dimensions, better balancing optimization across the ViT and VAE components. On ImageNet, this yields up to 10.57 gFID gain over naive MSE loss without classifier-free guidance. Twins also performs competitively on multimodal understanding benchmarks and improves reconstruction fidelity, narrowing the gap between understanding- and generation-oriented representations.

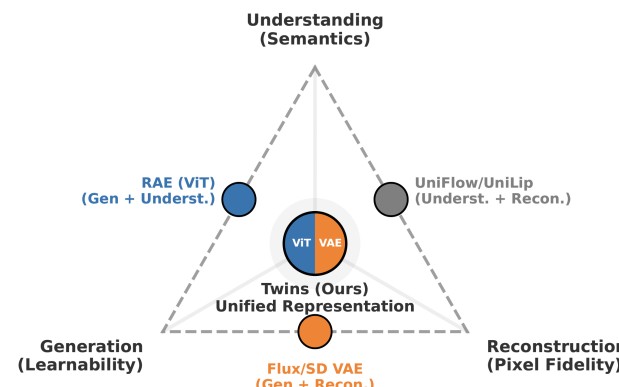

*Figure 1.* **Breaking the "Impossible Triangle" of visual tokenization.** Existing approaches (edges) are forced to trade off between Understanding, Reconstruction, and Generation. Twins (Ours) constructs a unified representation that explicitly fuses semantic-rich ViT features with detail-preserving VAE features, simultaneously satisfying all three objectives.

## 1. Introduction

Unified multimodal models (Zhou et al., 2024; Shi et al., 2024; Wu et al., 2025a; Chen et al., 2025e) have recently attracted increasing attention, motivated by the prospect of using a single model and a shared representation space to support both multimodal understanding and multimodal generation. Existing efforts can be broadly grouped into two routes: discrete (Wang et al., 2024c; Ma et al., 2025; Xie et al., 2024b) and continuous visual representations (Deng et al., 2025; Zhou et al., 2024). Discrete approaches first encode an image into a sequence of discrete tokens (Yu et al., 2021; Esser et al., 2020; Han et al., 2025), enabling both understanding and generation to operate on the same codebook: understanding leverages visual tokens for reasoning, while generation predicts in the same token space and then decodes tokens back to images. This shared token space naturally couples the two directions.

In contrast, unified models based on continuous visual representations (Deng et al., 2025; Zhou et al., 2024) often adopt a dual-representation design: a ViT feature space for understanding that is highly discriminative and captures rich semantics (*e.g.*, SigLIP2 (Tschannen et al., 2025)), alongside a VAE latent space for generation that prioritizes reconstruction fidelity (Yao et al., 2025; Kingma et al., 2019;

Work done during Kaixiong Gong's internship at Tencent Hunyuan. [*] indicates equal contribution. [†] indicates project leader. [1]The Chinese University of Hong Kong [2]Tencent, Hunyuan [3]City University of Hong Kong [4]Xiamen University [5]The Chinese University of Hong Kong, Shenzhen. Correspondence to: Xiangyu Yue <xyyue@ie.cuhk.edu.hk>.

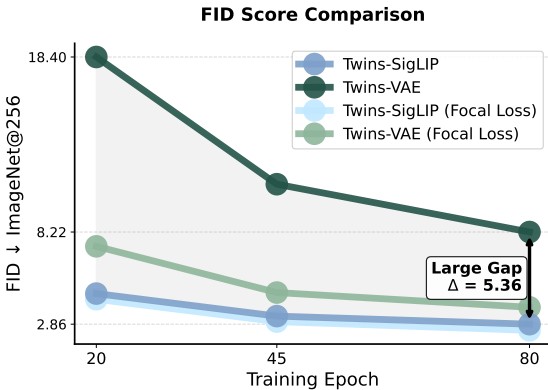

*Figure 2.* FID trajectories of the SigLIP and VAE components within the Twins representation. In the baseline setting, a critical failure mode emerges where the DiT fits SigLIP features well but struggles to model VAE latents. Focal Loss significantly mitigates this imbalance, leading to a substantial reduction in VAE FID ($\Delta = 5.36$ at 80 epochs) while maintaining performance on the SigLIP component.

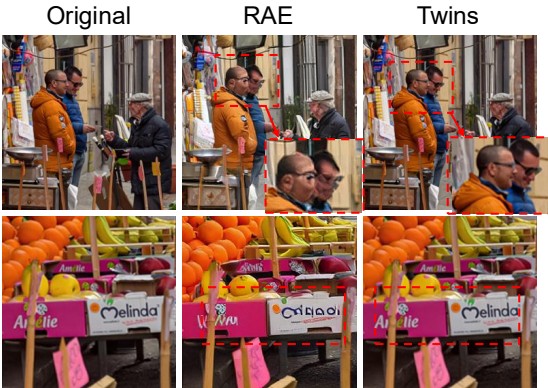

*Figure 3.* Reconstruction: RAE vs. Twins. RAE (DINOv2-B) fails to reconstruct the high-frequency and fine-grained details of original images.

Labs, 2024). This split reflects complementary inductive biases: ViT features are effective for semantic reasoning but tend to discard fine-grained pixel details, whereas VAE latents retain appearance details but provide weaker semantic separability. However, operating in two mismatched spaces comes with two key drawbacks. First, to "understand" its own generations, the system must perform an extra decode–encode round trip (latent $\rightarrow$ pixels $\rightarrow$ ViT features), increasing both computation and engineering complexity. Second, the mismatch breaks representational consistency, limiting the reuse of learned visual abstractions across understanding and generation. By contrast, language models (Brown et al., 2020; Achiam et al., 2023) both comprehend and generate in the same token space, yielding a coherent interface. Therefore, continuous approaches still lag behind discrete token spaces in terms of representation unification.

Prior efforts such as UniFlow (Yue et al., 2025) and UniLip (Tang et al., 2025) move toward unification by fine-tuning a CLIP-based encoder (Radford et al., 2021) to improve its reconstruction quality. However, to make generation easier, they compress the original high-dimensional CLIP features into a much lower-dimensional latent space (*e.g.*, 32 dimensions in UniLip and 64 in UniFlow). This dimensionality reduction, while beneficial for generation, can compromise the representational capacity for understanding, and the representation gap still exists. More recently, RAE (Zheng et al., 2025b) demonstrates that directly generating high-dimensional latents is feasible, exemplified by DINOv2 features (Oquab et al., 2023). Yet, the semantics-oriented embeddings are not designed for high-fidelity reconstruction, leading to noticeably poorer reconstruction quality (*e.g.*, a PSNR of 18.83 and visualization in Fig. 3). Taken together, existing approaches face a persistent tension: it remains challenging to obtain a single representation that

simultaneously supports strong understanding, high-quality reconstruction, and generation-friendly predictability—an apparent "impossible triangle" as illustrated in Fig. 1.

We propose a simple and efficient unified token space, **Twins**, by channel-wise concatenating semantic-rich ViT features from SigLIP2 (Tschannen et al., 2025) with detail-preserving VAE latents from FLUX.2 (Labs, 2024). Because the two components share the same token grid, Twins keeps the sequence length unchanged, and thus does not increase the quadratic attention cost with respect to token count. However, when training a Diffusion Transformer (DiT) to predict Twins, we observe a pronounced *optimization imbalance*: DiT fits the SigLIP2 component well but struggles to model the VAE component, as evidenced by the component-wise FID trajectories in Fig. 2.

We analyze this imbalance in Sec. 2 and attribute it to three sources of heterogeneity between the two spaces: spectral characteristics, intrinsic dimensionality, and condition-aligned vs. condition-independent uncertainty. To mitigate the model's undesired preference, we adopt a *focal* reweighting strategy for flow matching inspired by Focal Loss (Lin et al., 2017), upweighting hard residuals on the VAE channels. This simple calibration substantially improves VAE modeling (Fig. 2) while maintaining the SigLIP2 component, enabling DiT to jointly predict both semantic and fine-grained latents in a single representation. We further validate Twins on multimodal understanding and reconstruction benchmarks.

Our contributions are summarized as follows:

- We propose a simple channel-wise concatenation of ViT and VAE features to form a shared continuous representation (Twins) that supports both understanding and generation, while keeping the token length unchanged (hence no increase in quadratic attention cost).
- We identify a pronounced optimization imbalance when

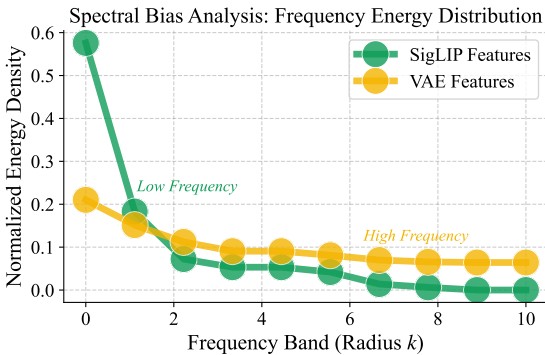

*Figure 4.* **Frequency energy distribution**. Normalized energy $e(k)$ across spatial frequency bands for SigLIP and Flux VAE. SigLIP features are dominated by low-frequency components, whereas Flux VAE retains significantly more energy in the high-frequency bands, indicating a richer preservation of spatial details.

training a DiT to predict Twins, and provide a systematic analysis of its causes.

- We adapt a focal reweighting strategy (inspired by Focal Loss) to flow matching on the VAE channels, effectively mitigating the imbalance during Twins modeling.
- We demonstrate substantial generation improvements over MSE, and show that Twins achieves comparable or better understanding performance than a strong single-encoder baseline (SigLIP2), with notable gains on fine-grained reconstruction enabled by richer visual detail.

## 2. Why DiT Prioritizes ViT over VAE?

Diffusion Transformers (DiTs) (Peebles & Xie, 2023) can model either ViT features (Zheng et al., 2025b) or VAE latents (Labs, 2024) when each is learned alone; however, learning them *jointly* in a shared representation is surprisingly non-trivial. We observe a consistent imbalance: DiT quickly captures the ViT component yet underfits the VAE component, resulting in poor FID scores and blurred, low-quality images as shown in Fig. 8. We attribute this *optimization imbalance* to three fundamental discrepancies between the two kinds of feature spaces: spectral characteristics, intrinsic dimensionality, and conditional dependency.

**1. Spectral Bias and Priority.** First, we analyze the signal frequency characteristics using Fast Fourier Transform (FFT). As shown in Fig. 4, the radial power spectrum reveals a distinct contrast: SigLIP features exhibit rapid spectral decay, indicating they are dominantly **low-frequency signals**. In contrast, VAE features maintain significant energy across high frequencies, behaving as broadband signals rich in texture and noise. According to the *Spectral Bias* theory (Rahaman et al., 2019), neural networks naturally prioritize learning low-frequency functions. Consequently, in the early training stages, the DiT network is inherently

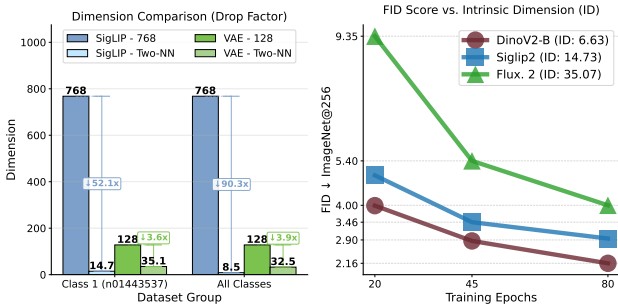

*Figure 5.* (Left) **Comparison of physical and intrinsic dimensions (ID) estimated via Two-NN.** A *dimensionality paradox* is observed: SigLIP has a higher physical dimension but a significantly lower ID than VAE, indicating a highly compressed manifold. (Right): **Generation performance (FID) across training epochs for features with varying IDs.** Features with higher ID (*e.g.*, Flux.2) exhibit significantly higher FID and slower convergence compared to those with lower ID (*e.g.*, DINOv2-B).

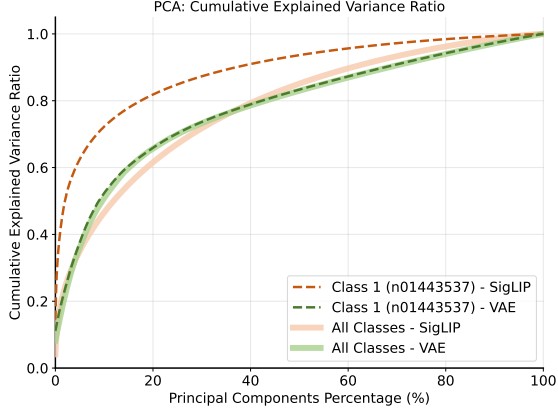

*Figure 6.* **PCA cumulative explained variance**. The drastic collapse of SigLIP's effective dimension under single-class conditions (dashed line) reveals its strong *conditional dependency*, whereas the consistently high dimensionality of VAE features indicates significant condition-independent uncertainty.

biased to fit the smooth SigLIP features first, while the high-frequency components of the VAE are perceived as difficult noise, delaying their optimization.

**2. Intrinsic Dimensionality and Learnability.** Second, we quantify the complexity of the feature manifolds using Two-Nearest Neighbors (Two-NN) estimation (Facco et al., 2017). We observe a *dimensionality paradox*: despite SigLIP having a much higher physical dimension ($D = 768$) than the VAE ($D = 128$), its class intrinsic dimension (ID) is significantly lower ($\text{ID}_{\text{SigLIP}} \approx 15$ vs. $\text{ID}_{\text{VAE}} \approx 35$), as illustrated in Figure 5 (left). Notably, we find that SigLIP's global ID ($\approx 8.5$) is lower than its single-class ID ($\approx 14.7$). This occurs because contrastive learning collapses classes into dense "semantic islands" that mis-

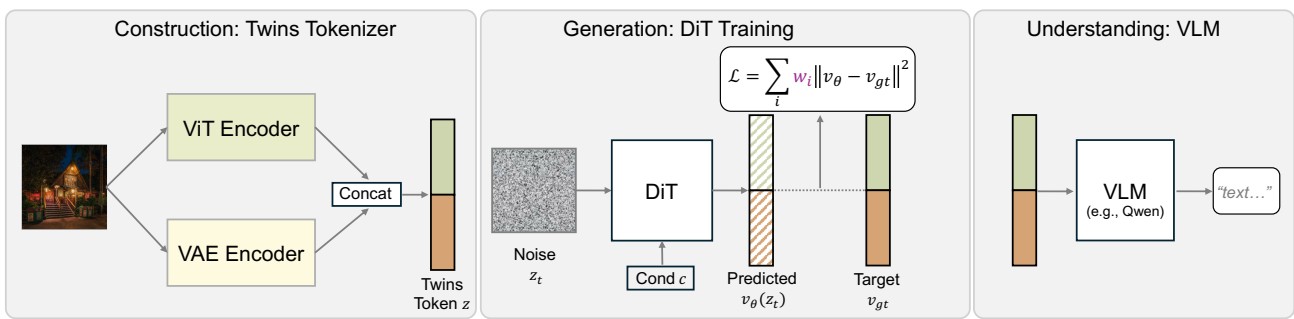

*Figure 7.* **Overview**: (Left) Construction: The Twins Tokenizer creates a unified visual token by concatenating semantic features from the SigLIP encoder and fine-grained latents from the Flux.2 VAE encoder along the channel dimension. (Middle) Generation: During DiT training, we employ Focal Loss within the Flow Matching objective to balance the learning of heterogeneous feature spaces. (Right) Understanding: For VLM inference, the unified Twins tokens are fed into a Large Language Model (*e.g.*, Qwen) via a projector to enable multimodal understanding tasks.

lead global estimation due to extreme inter-class separation. For conditional DiTs, the single-class ID serves as a more **faithful proxy** for optimization difficulty, as it reveals the intra-class variations that the model must accurately capture. This indicates that SigLIP features lie on a highly compressed manifold, whereas VAE features exhibit high entropy and structural complexity. As prior work (Pope et al., 2021) suggests that sample complexity scales with intrinsic dimension, the low-ID SigLIP manifold is significantly **easier to learn**, whereas the high-ID VAE manifold presents a more challenging optimization landscape. This is empirically verified by the strong correlation between ID and FID shown in Fig. 5(right).

**3. Conditional Alignment and Structural Dependency.** Finally, PCA analysis reveals the fundamental difference in how these features interact with generation conditions (*e.g.*, class labels $c$). As shown in Fig. 6, the effective dimension of SigLIP collapses drastically under single-class conditions compared to the global distribution. This implies the feature space is **highly structured**: given a condition $c$, the target feature is largely deterministic and confined to a low-dimensional subspace. Conversely, the dimensionality of VAE features remains high under conditioning. This suggests that VAE features contain significant uncertainty that is statistically independent of $c$.

Together, these factors induce a clear **optimization imbalance**. The network prefers to optimize the low-frequency, low-intrinsic-dimension, and condition-aligned SigLIP objective. Meanwhile, the VAE objective, being high-frequency, high-intrinsic-dimension, and conditionally unpredictable, is overlooked by the model. Under the Mean Squared Error (MSE) loss, the network struggles to capture the high-frequency details of the VAE features, causing the optimization to stagnate in a local optimum. Therefore, we introduce Focal Loss for Flow Matching to remedy the optimization imbalance.

## 3. Method

### 3.1. Preliminaries: Flow Matching

For image generation, we adopt the Flow Matching framework. Let $\boldsymbol{x}_0 \sim X_0$ denote a sample from the real data distribution, and let $\boldsymbol{x}_1 \sim X_1$ denote a noise sample drawn from a Gaussian distribution. Following Rectified Flow (Liu, 2022) and recent high-performing image generation models (Esser et al., 2024; Labs, 2024), we construct the intermediate corrupted sample at time $t \in [0, 1]$ via linear interpolation:

$$\boldsymbol{x}_t = (1 - t)\boldsymbol{x}_0 + t\boldsymbol{x}_1, \quad t \in [0, 1]. \tag{1}$$

We then train a transformer-based network $\boldsymbol{v}_\theta(\boldsymbol{x}_t, t)$ (Peebles & Xie, 2023; Ma et al., 2024) to estimate the corresponding velocity field using a mean squared error (MSE) objective:

$$\mathcal{L}(\theta) = \mathbb{E}_{t, \boldsymbol{x}_0, \boldsymbol{x}_1} \left[ \|\boldsymbol{v} - \boldsymbol{v}_\theta(\boldsymbol{x}_t, t)\|^2 \right], \tag{2}$$

where the target velocity is defined as $\boldsymbol{v} = \boldsymbol{x}_1 - \boldsymbol{x}_0$.

### 3.2. Twins: Unified Representation

We construct a unified embedding by concatenating features from two pretrained encoders: a SigLIP2 (Tschannen et al., 2025) ViT and a Flux.2 VAE (Labs, 2024). We employ the Flux.2 VAE as it is a widely-adopted tokenizer for high-fidelity image generation. These two encoders serve distinct yet **complementary roles**. The ViT, aligned with the language modality, excels at extracting high-level *semantic* representations essential for understanding content. In contrast, the VAE is optimized for pixel-level reconstruction, capturing fine-grained *low-level* details (*e.g.*, texture and local structure) that purely semantic embeddings often miss. By integrating them, our unified embedding combines rich semantic alignment with high-fidelity visual preservation.

Formally, let $I \in \mathbb{R}^{3 \times H \times W}$ denote an image of height $H$ and width $W$. We employ a ViT encoder $f_{\text{vit}}$ and a

VAE encoder $f_{\text{vae}}$ that share the same patch size $P$, yielding the same number of tokens $L = \frac{H}{P} \cdot \frac{W}{P}$ (assuming $H$ and $W$ are divisible by $P$). Denote $f_{\text{vit}}(I) \in \mathbb{R}^{L \times d_{\text{vit}}}$ and $f_{\text{vae}}(I) \in \mathbb{R}^{L \times d_{\text{vae}}}$ as the corresponding token embeddings. We construct the unified embedding by concatenating them along the channel dimension:

$$\boldsymbol{z} = \big[ f_{\text{vit}}(I), f_{\text{vae}}(I) \big], \qquad (3)$$

where $[\cdot, \cdot]$ denotes channel-wise concatenation. Consequently, $\boldsymbol{z} \in \mathbb{R}^{L \times (d_{\text{vit}} + d_{\text{vae}})}$. We perform fusion in the channel dimension rather than the sequence dimension, since increasing the sequence length would introduce additional overhead due to the $\mathcal{O}(L^2)$ complexity of attention with respect to the number of tokens $L$.

For image understanding, we substitute the original ViT embeddings with our unified embeddings. For image generation, we treat the unified embeddings as samples from the real data distribution.

### 3.3. Substitute MSE with Focal Loss

As aforementioned, modeling **Twins** representation is non-trivial. The DiT model favors SigLIP embeddings over VAE latents, for which we provide an analysis in Section 2. To prevent the model from falling into a suboptimal convergence where it prioritizes the more readily learnable semantic features (SigLIP) at the expense of structural details (VAE), we propose a feature-level Focal Loss (Lin et al., 2017). This loss re-weights the VAE regression task by assigning higher penalties to hard-to-learn features, effectively steering the optimization away from the semantic-only local optimum.

Let $\boldsymbol{v}_\theta(\boldsymbol{z}, t)$ denote the model prediction; $\boldsymbol{D}$ denote the set of VAE dimension indices; and $\boldsymbol{v}$ denote the ground-truth velocity, the original MSE loss on VAE dimensions has the form of:

$$\mathcal{L}_{\text{mse}} = \frac{1}{d_{vae}} \sum_{i \in \boldsymbol{D}} (\boldsymbol{v}_i - \boldsymbol{v}_\theta(\boldsymbol{z}, t)_i)^2. \qquad (4)$$

To strengthen the importance of difficult channels, we introduce a weighting scheme:

$$w_i = |\boldsymbol{v}_i - \boldsymbol{v}_\theta(\boldsymbol{z}, t)_i|^{2\gamma}, \qquad (5)$$

which is injected into the MSE loss to form:

$$\mathcal{L} = \frac{1}{d_{vae}} \sum_{i \in \boldsymbol{D}} w_i (\boldsymbol{v}_i - \boldsymbol{v}_\theta(\boldsymbol{z}, t)_i)^2. \qquad (6)$$

The $\gamma$ is set to $0.5$ in our experiments.

## 4. Experiment

In this section, we evaluate the proposed Twins unified embedding on reconstruction, multimodal understanding benchmark, and image generation. Detailed settings are depicted in corresponding subsections.

*Table 1.* **Results of reconstruction metrics on the $256 \times 256$ ImageNet-1K validation set.** "Ratio" denotes downsampling ratio; "Enc.-Dec." shows the types of encoder and decoder.

| Method | Enc.-Dec. | Ratio | PSNR ↑ | SSIM ↑ | rFID ↓ |
|---|---|---|---|---|---|
| *Generative Only Tokenizer* | | | | | |
| Cosmos-DI (Agarwal et al., 2025) | Discrete-Pixel | 16 | 19.98 | 0.54 | 4.40 |
| LlamaGen (Sun et al., 2024a) | Discrete-Pixel | 16 | 20.65 | 0.54 | 2.47 |
| Open-MAGVIT2 (Luo et al., 2024) | Discrete-Pixel | 16 | 22.70 | 0.64 | 1.67 |
| BSQ-ViT (Yang et al., 2021) | Discrete-Pixel | 16 | 28.14 | 0.81 | 0.45 |
| SD-VAE 1.x (Rombach et al., 2022) | Continuous-Pixel | 8 | 23.54 | 0.68 | 1.22 |
| SD-VAE 2.x (Rombach et al., 2022) | Continuous-Pixel | 8 | 23.54 | 0.68 | 1.22 |
| OmniTokenizer (Wang et al., 2024a) | Continuous-Pixel | 8 | 26.74 | 0.82 | 1.02 |
| SD-VAE XL (Podell et al., 2023) | Continuous-Pixel | 8 | 27.37 | 0.78 | 0.67 |
| Qwen-Image (Wu et al., 2025b) | Continuous-Pixel | 8 | 32.18 | 0.90 | 1.46 |
| SD-VAE 3 (Esser et al., 2024) | Continuous-Pixel | 8 | 31.29 | 0.87 | 0.20 |
| Wan2.1 (Wan et al., 2025a) | Continuous-Pixel | 8 | 31.34 | 0.89 | 0.95 |
| FLUX.1-VAE (Labs, 2024) | Continuous-Pixel | 8 | 32.74 | 0.92 | 0.18 |
| Cosmos-CI (Agarwal et al., 2025) | Continuous-Pixel | 16 | 25.07 | 0.70 | 0.96 |
| VA-VAE (Yao et al., 2025) | Continuous-Pixel | 16 | 27.96 | 0.79 | 0.28 |
| Wan2.2 (Wan et al., 2025b) | Continuous-Pixel | 16 | 31.25 | 0.88 | 0.75 |
| SelfTok (Luo et al., 2024) | Discrete-Diffusion | – | 24.14 | 0.71 | 0.70 |
| FlowMo-Hi (Shaulov et al., 2025) | Discrete-Diffusion | – | 26.93 | 0.79 | 0.56 |
| l-DeTok (Yang et al., 2025) | Continuous-Diffusion | 16 | – | – | 0.68 |
| *Unified Tokenizer* | | | | | |
| Show-o (Xie et al., 2024a) | Discrete-Pixel | 16 | 21.34 | 0.59 | 3.50 |
| QLIP-B (Zhao et al., 2025) | Discrete-Pixel | 16 | 23.16 | 0.63 | 3.21 |
| VILA-U (Wu et al., 2024b) | Discrete-Pixel | 16 | – | – | 1.80 |
| Tokenflow (Qu et al., 2025) | Discrete-Pixel | 16 | 21.41 | 0.69 | 1.37 |
| DualViTok (Huang et al., 2025) | Discrete-Pixel | 16 | 22.53 | 0.74 | 1.37 |
| DualToken (Song et al., 2025) | Discrete-Pixel | 16 | 23.56 | 0.74 | 0.54 |
| MUSE-VL (Xie et al., 2024b) | Discrete-Pixel | 16 | 20.14 | 0.65 | 2.26 |
| SemHiTok (Chen et al., 2025f) | Discrete-Pixel | 16 | – | – | 1.16 |
| UniTok (Ma et al., 2025) | Discrete-Pixel | 16 | 27.28 | 0.77 | 0.41 |
| SeTok (Wu et al., 2025c) | Discrete-Pixel | – | – | – | 2.07 |
| EMU2 (Sun et al., 2024c) | Continuous-Diffusion | 14 | 13.49 | 0.42 | 3.27 |
| BLIP3-o (Chen et al., 2025c) | Continuous-Diffusion | 16 | 14.71 | 0.58 | 3.18 |
| UniFlow (*SigLIP2*) (Yue et al., 2025) | Continuous-Diffusion | 16 | 29.38 | 0.93 | 0.62 |
| UniFlow (*DINOv2*) (Yue et al., 2025) | Continuous-Diffusion | 14 | 31.01 | 0.94 | 0.54 |
| UniFlow (*InternViT*) (Yue et al., 2025) | Continuous-Diffusion | 14 | 33.23 | 0.96 | 0.26 |
| UniLIP (Tang et al., 2025) | Continuous-Pixel | 32 | 22.99 | 0.75 | 0.79 |
| RAE (Zheng et al., 2025b) | Continuous-Pixel | 14 | 18.83 | 0.50 | 0.57 |
| Twins | Continuous-Pixel | 16 | 31.46 | 0.90 | 0.11 |

### 4.1. Reconstruction

A critical bottleneck for unified models, particularly those based on discrete tokens or semantic-only encoders, is the loss of visual information during encoding. We evaluate the reconstruction quality of Twins on the ImageNet-1K validation set (Deng et al., 2009) and compare it against state-of-the-art visual tokenizers.

**SOTA-Level Fidelity.** As presented in Table 1, Twins achieves state-of-the-art reconstruction performance with a PSNR of 31.46, SSIM of 0.90, and a rFID of 0.11.

**Solving the Semantic-Reconstruction Trade-off.** A key comparison is against RAE, which attempts to decode images directly from semantic features. RAE achieves a poor PSNR of 18.83 and a high rFID of 0.57, highlighting the difficulty of recovering pixel-level details from semantic embeddings. In contrast, Twins leverages the concatenated VAE features to handle high-frequency details while the ViT component handles semantics. This design allows Twins to match the reconstruction quality of dedicated generative autoencoders like Wan2.2 (31.25) and SD-VAE 3 (31.29). Twins ensures that the unified representation retains the pixel-perfect consistency required for high-quality generation and image editing.

*Table 2.* **Results on multimodal (image and text) benchmarks.**

| Method | Encoder | LLM | Res. | POPE | GQA | TQA | MMB | MME-S | MME-P |
|---|---|---|---|---|---|---|---|---|---|
| **Understanding Only MLLM** | | | | | | | | | |
| InstructBLIP (Dai et al., 2023) | CLIP-G | Vicuna-7B | 224 | – | 49.2 | 50.7 | – | – | – |
| MiniGPT-4 (Zhu et al., 2023) | CLIP-G | Vicuna-13B | 224 | – | – | – | – | 1158.7 | 866.6 |
| InstructBLIP (Dai et al., 2023) | CLIP-G | Vicuna-13B | 224 | 78.9 | 49.5 | 50.7 | 36.0 | – | 1212.8 |
| IDEFICS (Laurençon et al., 2024) | CLIP-H | LLaMA-7B | 224 | – | 38.4 | 25.9 | 48.2 | – | – |
| mPLUG-Owl2 (Ye et al., 2024) | CLIP-L | LLaMA-2-7B | 448 | 86.2 | 56.1 | 58.2 | 64.5 | – | – |
| InternVL-Chat (Chen et al., 2024) | InternViT-6B | Vicuna-7B | 224 | 85.2 | 57.7 | – | – | – | 1298.5 |
| LLaVA-1.5 (Liu et al., 2023) | CLIP-L | Vicuna-7B | 336 | 85.9 | 62.0 | 46.1 | 64.3 | – | 1510.7 |
| Qwen-VL-Chat (Wang et al., 2024b) | CLIP-G | Qwen-7B | 448 | – | 57.5 | – | – | 1848.3 | 1487.5 |
| LLaVA-OneVision (Li et al., 2024a) | SigLiP-SO400M | Qwen-2-7B | 384 | – | – | 46.1 | 80.8 | 1998.0 | 1580.0 |
| **Unified MLLM** | | | | | | | | | |
| DreamLLM (Dong et al., 2023) | CLIP-L | Vicuna-7B | 224 | – | – | 41.8 | – | – | – |
| LaVIT (Liu et al., 2024a) | CLIP-G | LLaMA-2-7B | 224 | – | 48.0 | – | 58.0 | – | – |
| Unified-IO 2 (Lu et al., 2023) | VQ-GAN | 6.8B from scratch | 384 | 87.7 | 59.1 | – | 71.5 | 1338.0 | – |
| Janus (Wu et al., 2025a) | SigLIP-L | DeepSeek-LLM-1.3B | 384 | 87.0 | 59.1 | – | 69.4 | – | 1338.0 |
| LWM (Liu et al., 2024c) | VQ-GAN | LLaMA-2-7B | 256 | 75.2 | 44.8 | 18.8 | – | – | – |
| SEED-X (Ge et al., 2024) | Qwen-VL-ViT | LLaMA-2-13B | 448 | 84.2 | 47.9 | – | – | – | 1435.7 |
| Show-o (Xie et al., 2024a) | MAGVIT-v2 | Phi-1.5-1.3B | 512 | 80.0 | 58.0 | – | – | – | 1097.2 |
| MetaMorph (Gupta et al., 2022) | SigLIP-SO400M | LLaMA-3.1-8B | 384 | – | – | 60.5 | 75.2 | – | – |
| Orthus (Kou et al., 2024) | VAE | Chameleon-7B | 256 | 79.6 | 52.8 | – | – | – | 1265.8 |
| SynerGen-VL (Li et al., 2025) | SBER-MoVQ-GAN | InternLM2-MoE-2.4B | 512 | 85.3 | 59.7 | – | 53.7 | – | 1381.0 |
| Liquid (Wu et al., 2024a) | VQ-GAN | Gemma-7B | 512 | 81.1 | 58.4 | 42.4 | – | – | 1119.0 |
| VILA-U (Lin et al., 2024) | SigLIP-SO400M | LLaMA-2-7B | 384 | 85.8 | 60.8 | 60.8 | – | – | 1401.8 |
| Janus-Pro (Chen et al., 2025e) | SigLIP-L | DeepSeek-LLM-7B | 384 | 87.4 | 62.0 | – | 79.2 | – | 1567.1 |
| Show-o2 (Xie et al., 2025) | Wan2.1-VAE+ViT-SO400M | Qwen2.5-7B | 432 | – | 63.1 | – | 79.3 | – | 1620.5 |
| **MLLM with Unified Tokenizer** | | | | | | | | | |
| VILA-U (Wu et al., 2024b) | SigLIP-SO400M | Vicuna-7B | 256 | 81.6 | – | – | – | – | 1311.6 |
| UniTok (Ma et al., 2025) | Vitamin-L | Vicuna-7B | 256 | 81.7 | – | – | – | – | 1448.0 |
| SemHiTok (Chen et al., 2025f) | SigLIP-L | Vicuna-7B | 256 | 84.2 | 61.0 | – | 60.3 | – | 1400.6 |
| QLIP (Zhao et al., 2025) | CLIP-L | Vicuna-7B | 392 | 86.1 | 61.8 | 55.2 | – | – | 1498.3 |
| TokenFlow-B (Qu et al., 2025) | CLIP-B | Vicuna-13B | 224 | 84.0 | 59.3 | 49.8 | 55.3 | 1660.4 | 1353.6 |
| TokenFlow-L (Qu et al., 2025) | ViTamin-XL | Vicuna-13B | 256 | 85.0 | 60.3 | 54.1 | 60.3 | 1622.9 | 1365.4 |
| UniTok (Ma et al., 2025) | Vitamin-L | LLaMA-2-7B | 256 | 83.2 | 61.1 | 51.6 | – | – | 1448.0 |
| TokLIP (Lin et al., 2025) | VQ-GAN+ViT-SO400M | Qwen2.5-7B | 384 | 84.1 | 59.5 | – | 67.6 | – | 1448.4 |
| TokenFlow-XL (Qu et al., 2025) | SigLIP-SO400M | Qwen2.5-14B | 384 | 87.8 | 62.5 | 62.3 | 76.8 | 1922.2 | 1551.1 |
| UniFlow (Yue et al., 2025) | SigLIP2-SO400M | Vicuna-7B | 256 | 87.94 | 63.29 | 58.0 | 68.38 | 1823.0 | 1477.9 |
| UniFlow (Yue et al., 2025) | InternViT-300M | Vicuna-7B | 448 | 88.97 | 63.35 | 61.85 | 67.10 | 1803.0 | 1505.1 |
| **Twins** | SigLIP2-SO400M | Qwen2.5-7B | 384 | 87.15 | 64.54 | 56.92 | 77.00 | 1826.8 | 1512.1 |
| **Twins** | SigLIP2-SO400M + Flux.2 VAE | Qwen2.5-7B | 384 | 87.82 | 64.93 | 58.89 | 77.00 | 1971.0 | 1588.8 |

## 4.2. Multimodal Understanding Results

To evaluate the representation capability of Twins for multimodal understanding, we integrate our unified representation with a LLaVA-style (Liu et al., 2024b) pipeline. Specifically, we replace the standard vision encoder with our Twins encoder and train a VLM using Qwen2.5-7B (Qwen et al., 2025) as the language backbone. We use LLaVA-558k (Liu et al., 2024b) for pretraining and Cambrian-737k (Tong et al., 2024) for instruction fine-tuning, with all training settings consistent with LLaVA-1.5 (Liu et al., 2024b). We set up an important baseline of SigLIP2 (Tschannen et al., 2025), as Twins is composed of a SigLIP2 ViT feature and a Flux.2 VAE feature.

**Competitive Performance with Specialized Encoders.** Table 2 shows that Twins generally outperforms the strong baseline SigLIP2 encoder. Moreover, we observe that the inclusion of low-level features yields improvements across several fine-grained tasks, such as GQA (64.93 vs. 64.54) and TQA (58.89 vs. 56.92). We attribute this to the fact that Twins' features preserve low-level visual details (*e.g.*, texture, exact shape) that are often abstracted away by high-level semantic encoders, thereby enriching the visual information available to the LLM.

## 4.3. Image Generation Results

**Setting**: Twins concatenates the features of a SigLIP2-B (Tschannen et al., 2025) (following RAE (Zheng et al., 2025b)) and a Flux.2 VAE (Labs, 2024). We employ the Flux.2 VAE as it is a widely-adopted tokenizer for high-fidelity image synthesis. Successfully modeling such a sophisticated latent space demonstrates the practical applicability of our approach. We also set up another baseline of SigLIP2 to manifest that DiT can learn semantic embed-

*Table 3.* **Class-conditional performance on ImageNet 256×256.** Baseline Flux.2 VAE and SigLIP2 indicate that DiT is trained in the Flux.2 VAE latents or SigLIP2 latents only. Twins methods below Flux.2 VAE are decoded with VAE decoder while others are decoded with SigLIP2 decoder.

| Method | Epochs | PSNR | Generation@256 w/o guidance | | | | Generation@256 w/ guidance | | | |
|---|---|---|---|---|---|---|---|---|---|---|
| | | | gFID↓ | IS↑ | Prec.↑ | Rec.↑ | gFID↓ | IS↑ | Prec.↑ | Rec.↑ |
| *Latent Diffusion with Semantic Embedding (Low PSNR)* | | | | | | | | | | |
| RAE (DINOv2-B, DiT$^{DH}$) (Zheng et al., 2025b) | 20 | 18.83 | 3.71 | 198.7 | 0.86 | 0.50 | – | – | – | – |
| | 80 | | 2.16 | 214.8 | 0.82 | 0.59 | – | – | – | – |
| | 800 | | **1.51** | **242.9** | 0.79 | 0.63 | **1.13** | 262.6 | 0.78 | **0.67** |
| *Latent Diffusion with Unified Embedding (Ours, High-Dimensional, High PSNR)* | | | | | | | | | | |
| Baseline: Flux.2 VAE (Labs, 2024) | 20 | 31.46 | 9.35 | 101.28 | 0.71 | 0.61 | - | - | - | - |
| | 80 | | 3.99 | 157.77 | 0.74 | 0.66 | 3.06 | 321.87 | 0.86 | 0.54 |
| Baseline: Twins MSE Loss | 20 | 31.46 | 23.69 | 77.03 | 0.57 | 0.52 | - | - | - | - |
| | 80 | | 14.41 | 112.98 | 0.62 | 0.59 | - | - | - | - |
| **Twins, Focal Loss** | 20 | 31.46 | 7.38 | 140.31 | 0.74 | 0.55 | - | - | - | - |
| | 80 | | 3.84 | 184.06 | 0.75 | 0.59 | 1.59 | 245.06 | 0.76 | 0.64 |
| Baseline: SigLIP2 (Tschannen et al., 2025) | 20 | 19.11 | 4.97 | 167.56 | 0.81 | 0.51 | - | - | - | - |
| | 80 | | 2.94 | 193.91 | **0.89** | 0.59 | 1.84 | 215.54 | 0.79 | 0.62 |
| Baseline: Twins MSE Loss | 20 | 31.46 | 4.64 | 162.15 | 0.82 | 0.53 | - | - | - | - |
| | 80 | | 2.86 | 185.28 | 0.79 | 0.59 | - | - | - | - |
| **Twins, Focal Loss** | 20 | 31.46 | 4.31 | 172.82 | 0.84 | 0.52 | - | - | - | - |
| | 80 | | 2.50 | 205.38 | 0.81 | 0.57 | 1.47 | 248.95 | 0.80 | 0.63 |

*Table 4.* **Class-conditional performance on ImageNet 512×512.**

| Method | Epoch | Generation@512 | | | |
|---|---|---|---|---|---|
| | | gFID↓ | IS↑ | Prec.↑ | Rec.↑ |
| RAE (Zheng et al., 2025b) | 400 | 1.13 | 259.6 | 0.80 | 0.63 |
| *Ours (w/o guidance)* | | | | | |
| Baseline: Flux.2 VAE (Labs, 2024) | 80 | 4.57 | 152.88 | 0.80 | 0.65 |
| Baseline: Twins, MSE Loss | 80 | 6.80 | 153.18 | 0.78 | 0.60 |
| **Twins, Focal Loss** | 80 | 3.78 | 187.85 | 0.80 | 0.59 |
| *Ours (w/ guidance)* | | | | | |
| **Twins, Focal Loss** | 80 | 1.79 | 237.36 | 0.80 | 0.63 |

ding and detail embedding well simultaneously with our proposed Focal Loss. Following this, we adopt the DDT head design from DDT (Wang et al., 2025) and use auto-guidance (Karras et al., 2025) as the guidance method. All our experiments are under the same network architecture, learning rate, noise shift, and other hyperparameters.

Table 3 reports the generation metrics on ImageNet@256. In the *w/o guidance* setting, we observe that the model trained with default MSE loss yields a significantly worse FID compared to the Flux.2 VAE baseline. This degradation highlights the **inadequacy of the MSE objective** in jointly optimizing heterogeneous modalities, as it tends to neglect the high-entropy VAE features in favor of the easier SigLIP ones. In contrast, our proposed Focal Loss successfully mitigate this issue, substantially improving the prediction

accuracy for Flux.2 VAE features. Interestingly, we also observe that this adjustment leads to slight improvements in SigLIP2 feature prediction. We attribute this phenomenon to the **gradient balancing effect** of Focal Loss. By down-weighting the loss contribution from the well-converged SigLIP features, the objective suppresses the dominance of "easy" gradients, preventing the optimization from stagnating in local optima. Furthermore, the successful modeling of fine-grained VAE details likely forces the shared backbone to learn more robust and multi-scale representations, which in turn benefits the semantic alignment of SigLIP. Finally, with classifier-free guidance, Twins achieves a strong FID of 1.59. Although RAE (Zheng et al., 2025b) achieves a slightly lower FID, Twins outperforms it in terms of reconstruction quality, achieving a significantly higher PSNR.

Additional results on the ImageNet@512 dataset are presented in Table 4. We observe that the performance trends remain **consistent with** those in the 256 × 256 resolution, reinforcing the validity of our analysis across different scales.

## 5. Related Work

### 5.1. Visual Tokenizer for Generative Modeling

Visual tokenizers compress images into compact latent representations to facilitate efficient generation. Early paradigms relied on continuous VAEs (Kingma et al., 2019; Rombach et al., 2022) or discrete VQ-VAEs (Van Den Oord et al., 2017; Yu et al., 2021), often yielding suboptimal reconstruction fidelity. To improve image quality, recent continuous

models like FLUX (Labs, 2024) and SD3 (Esser et al., 2024) significantly expand latent channel dimensions, while discrete approaches like MagVIT-v2 (Yu et al., 2024a) enhance codebook utilization. Conversely, to improve efficiency by compressing the sequence length of visual tokens. In the continuous domain, the DC-AE series (Chen et al., 2025b;d) achieves high compression rates (e.g., $64\times$) while maintaining perceptual quality. In the discrete domain, 1D tokenizers such as TiTok (Yu et al., 2024b) and FlexTok (Bachmann et al., 2025) effectively map 2D image grids into compact 1D sequences, significantly reducing the computational burden for autoregressive modeling.

Despite these advances, recent studies have identified a fundamental trade-off between reconstruction fidelity and generative capability in tokenizers trained solely with reconstruction objectives (Yao et al., 2025). Addressing this, recent works such as VA-VAE (Yao et al., 2025), VFM-Tok (Zheng et al., 2025a), and MAETok (Chen et al., 2025a) integrate semantic guidance from vision foundation models to enrich the latent space for better generation.

### 5.2. Unified Tokenizer for Understanding and Generation

Developing a single tokenizer for both understanding and generation remains a core challenge due to the inherent conflict between high-level semantic abstraction and low-level pixel fidelity (Fan et al., 2025).

To bridge this gap, discrete approaches like UniTok (Ma et al., 2025) and QLIP (Zhao et al., 2025) align quantization codebooks with semantic concepts, often incurring information loss compared to continuous features. Continuous methods follow two paradigms: 1) *Unified training*, where models like VILA-U (Wu et al., 2024b) and UniFlow (Yue et al., 2025) employ joint objectives or self-distillation to fuse capabilities; 2) *Repurposing representation models*, where RAE (Zheng et al., 2025b) decodes directly from frozen semantic encoders (e.g., SigLIP) but sacrifices reconstruction fidelity due to lost high-frequency details. While SVG (Shi et al., 2025) mitigates this via an additional residual encoder, it increases architectural complexity.

In contrast to approaches that require complex alignment or architectural redesign, we propose a minimalistic paradigm: directly integrating off-the-shelf tokenizers. We construct a unified continuous space by concatenating features from a Vision Transformer (ViT) (Tschannen et al., 2025), which captures semantics, with latents from a Variational Autoencoder (VAE) (Kingma et al., 2019), which ensure generative fidelity. This simple channel-wise concatenation (Twins) preserves the strengths of both representations, avoiding any additional information bottleneck in the latent space.

### 5.3. Unified Multimodal Models

The pursuit of unified multimodal models aims to handle both perception and generation tasks within a single transformer architecture. Early works like Chameleon (Team, 2024) and Emu3 (Wang et al., 2024c) tokenize images into discrete tokens and model them autoregressively alongside text tokens. Show-o (Xie et al., 2024a) and Show-o2 (Xie et al., 2025) further unify multimodal understanding and generation by integrating autoregressive and diffusion modeling into a single transformer.

On the continuous side, models like Janus (Wu et al., 2025a) and Janus-Pro (Chen et al., 2025e) typically decouple visual encoding, using separate encoders for different tasks within the same framework. While approaches like Dream-LLM (Dong et al., 2023) and SEED-X (Ge et al., 2024) explore interleaving generation and understanding objectives, they often rely on separate visual representations for input and output. Notably, Bagel (Deng et al., 2025) utilizes both ViT and VAE features for understanding but generates solely VAE latents. This asymmetry results in a disjoint representation space, effectively preventing the model from directly perceiving its own generations without additional re-encoding steps. In contrast, our work establishes a truly shared continuous representation that serves both understanding and generation simultaneously, allowing the model to operate within a single unified space without requiring extra encoding or decoding processes.

## 6. Conclusion

In this work, we introduced **Twins**, a unified representation that is training-free, semantically rich, and capable of high-fidelity generation by leveraging existing powerful encoders. While promising, we identified that jointly modeling these heterogeneous features is non-trivial, as standard DiT models tend to prioritize the easier ViT features.

Crucially, we provided a comprehensive analysis to demystify this failure mode. We attributed the training instability to an **optimization imbalance** driven by three fundamental discrepancies between the feature spaces: (1) **Spectral characteristics**, where the network favors low-frequency ViT signals; (2) **Intrinsic dimensionality**, where the low-ID ViT manifold is significantly easier to learn than the high-ID VAE space; and (3) **Conditional dependency**, where ViT features are structurally aligned with semantic conditions.

Guided by these insights, we propose **Focal Loss** for generation to calibrate the learning process. Our experiments demonstrate that this strategy successfully mitigates the gradient dominance of ViT features, enabling high-quality prediction of both modalities and establishing a new baseline for unified representation generation.

## Impact Statement

This paper presents work whose goal is to advance the field of machine learning. There are many potential societal consequences of our work, none of which we feel must be specifically highlighted here.

## Acknowledgement

This work is partially supported by the National Natural Science Foundation of China (No. 62306261), HK RGC-Early Career Scheme (No. 24211525), ITSP Platform Project (No. ITS/600/24FP) and the SHIAE Grant (No. 8115074). This study was supported in part by the Centre for Perceptual and Interactive Intelligence, a CUHK-led InnoCentre under the InnoHK initiative of the Innovation and Technology Commission of the Hong Kong Special Administrative Region Government. This work is also partially supported by Hong Kong RGC Strategic Topics Grant (No. STG1/E-403/24-N), and CUHK-CUHK(SZ)-GDST Joint Collaboration Fund (No. YSP26-4760949).

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

# A. Appendix

## A.1. Details of Intrinsic Dimension Estimation (Two-NN)

To quantify the complexity of the feature manifolds, we employ the Two-Nearest Neighbors (Two-NN) algorithm proposed by (Facco et al., 2017).

The Two-NN method is based on the statistics of the distances between each point and its first two nearest neighbors. Let $\{\mathbf{x}_i\}_{i=1}^N$ be a set of $N$ data points in a $D$-dimensional space. For each point $\mathbf{x}_i$, we calculate the distances to its first and second nearest neighbors, denoted as $r_{i,1}$ and $r_{i,2}$ respectively.

The core assumption is that, locally, the points are drawn from a Poisson process with constant density. Under this assumption, the ratio of the two distances:

$$\mu_i = \frac{r_{i,2}}{r_{i,1}}, \quad \mu_i \in [1, \infty) \tag{7}$$

follows a Pareto distribution. Specifically, the cumulative distribution function (CDF) of the ratio $\mu$ is given by:

$$F(\mu) = 1 - \mu^{-d} \tag{8}$$

where $d$ represents the intrinsic dimension of the manifold.

To estimate $d$ from a finite dataset, the following empirical steps are performed:

1. Compute Ratios: For each data point $i$, find the distances $r_{i,1}$ and $r_{i,2}$ and compute $\mu_i = r_{i,2}/r_{i,1}$.

2. Empirical Distribution: Sort the computed ratios $\{\mu_i\}$ in ascending order such that $\mu_{(1)} \leq \mu_{(2)} \leq \cdots \leq \mu_{(N)}$. For each $i$, the empirical CDF is estimated as $F(\mu_{(i)}) \approx \frac{i}{N}$.

3. Linear Regression: By taking the logarithm of both sides of Equation 8, we obtain a linear relationship:

$$\log(1 - F(\mu)) = -d \log(\mu) \tag{9}$$

   Technically, we define coordinates $X_i = \log(\mu_{(i)})$ and $Y_i = -\log(1 - \frac{i}{N})$. The intrinsic dimension $d$ is then estimated as the slope of the line $Y = dX$ passing through the origin, using a simple least-squares fit.

In Section 2, we applied Two-NN to the feature distributions of SigLIP and Flux VAE. The significant difference between the estimated $d$ and the physical dimension $D$ (*e.g.*, for SigLIP, $d \approx 15$ while $D = 768$) reveals that SigLIP features in fact lie in a low-dimensional distribution which is easier to learn compared with the high-ID VAE latents (as shown in Fig. 5).

## A.2. More Comparison Results

Due to the space limit, we relocate some comparison results from the main paper to here. Image generation on ImageNet@256 is in Table 5.

We show generation results comparison between MSE Loss and Focal Loss on Twins as shown in Fig. 8.

Twins Focal Loss

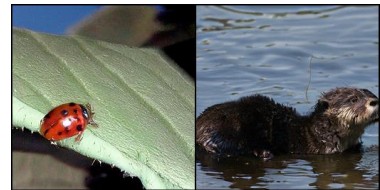

Twins MSE Loss

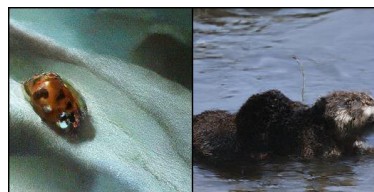

*Figure 8.* Generation: MSE vs. Focal. MSE Loss generates blurred and distorted images.

*Table 5.* **Class-conditional performance on ImageNet 256×256.**

| Method | Epochs | PSNR | Generation@256 w/o guidance | | | | Generation@256 w/ guidance | | | |
|---|---|---|---|---|---|---|---|---|---|---|
| | | | gFID↓ | IS↑ | Prec.↑ | Rec.↑ | gFID↓ | IS↑ | Prec.↑ | Rec.↑ |
| *Autoregressive* | | | | | | | | | | |
| VAR (Tian et al., 2024) | 350 | - | 1.92 | 323.1 | 0.82 | 0.59 | 1.73 | **350.2** | 0.82 | 0.60 |
| MAR (Li et al., 2024b) | 800 | - | 2.35 | 227.8 | 0.79 | 0.62 | 1.55 | 303.7 | 0.81 | 0.62 |
| xAR (Ren et al., 2025) | 800 | - | - | - | - | - | 1.24 | 301.6 | **0.83** | 0.64 |
| *Pixel Diffusion* | | | | | | | | | | |
| ADM (Dhariwal & Nichol, 2021) | 400 | - | 10.94 | 101.0 | 0.69 | 0.63 | 3.94 | 215.8 | **0.83** | 0.53 |
| RIN (Jabri et al., 2023) | 480 | - | 3.42 | 182.0 | - | - | - | - | - | - |
| *Latent Diffusion with VAE* | | | | | | | | | | |
| DiT (Peebles & Xie, 2023) | 1400 | - | 9.62 | 121.5 | 0.67 | 0.67 | 2.27 | 278.2 | **0.83** | 0.57 |
| MaskDiT (Zheng et al., 2024) | 1600 | - | 5.69 | 177.9 | 0.74 | 0.60 | 2.28 | 276.6 | 0.80 | 0.61 |
| VA-VAE (Yao et al., 2025) | 800 | - | 2.17 | 205.6 | 0.77 | 0.65 | 1.35 | 295.3 | 0.79 | 0.65 |
| REPA (Yu et al., 2025) | 800 | - | 5.78 | 158.3 | 0.70 | 0.68 | 1.29 | 306.3 | 0.79 | 0.64 |
| *Latent Diffusion with Semantic Embedding (Low PSNR)* | | | | | | | | | | |
| RAE (DiT$^{DH}$) | 800 | 18.83 | **1.51** | **242.9** | 0.79 | 0.63 | **1.13** | 262.6 | 0.78 | **0.67** |
| *Ours: Latent Diffusion with Unified Embedding (High PSNR)* | | | | | | | | | | |
| Baseline: Flux.2 VAE | 20 | 31.46 | 9.35 | 101.28 | 0.71 | 0.61 | - | - | - | - |
| | 80 | 31.46 | 3.99 | 157.77 | 0.74 | 0.66 | 3.06 | 321.87 | 0.86 | 0.54 |
| Baseline: Twins MSE | 20 | 31.46 | 23.69 | 77.03 | 0.57 | 0.52 | - | - | - | - |
| | 80 | 31.46 | 14.41 | 112.98 | 0.62 | 0.59 | - | - | - | - |
| **Twins, Focal Loss** | 20 | 31.46 | 7.38 | 140.31 | 0.74 | 0.55 | - | - | - | - |
| | 80 | 31.46 | 3.84 | 184.06 | 0.75 | 0.59 | 1.59 | 245.06 | 0.76 | 0.64 |
| Baseline: SigLIP2 | 20 | 19.11 | 4.97 | 167.56 | 0.81 | 0.51 | - | - | - | - |
| | 80 | 19.11 | 2.94 | 193.91 | **0.89** | 0.59 | 1.84 | 215.54 | 0.79 | 0.62 |
| **Twins, Focal Loss** | 20 | 31.46 | 4.31 | 172.82 | 0.84 | 0.52 | - | - | - | - |
| | 80 | 31.46 | **2.50** | **205.38** | 0.81 | 0.57 | **1.47** | **248.95** | 0.80 | **0.63** |

